# β-Galactosidase-Producing Isolates in Mucoromycota: Screening, Enzyme Production, and Applications for Functional Oligosaccharide Synthesis

**DOI:** 10.3390/jof7030229

**Published:** 2021-03-19

**Authors:** Bettina Volford, Mónika Varga, András Szekeres, Alexandra Kotogán, Gábor Nagy, Csaba Vágvölgyi, Tamás Papp, Miklós Takó

**Affiliations:** 1Department of Microbiology, Faculty of Science and Informatics, University of Szeged, Közép fasor 52, H-6726 Szeged, Hungary; bettina.volford86@gmail.com (B.V.); varga.j.monika@gmail.com (M.V.); szandras@bio.u-szeged.hu (A.S.); primula15@gmail.com (A.K.); nagygab86@gmail.com (G.N.); csaba@bio.u-szeged.hu (C.V.); pappt@bio.u-szeged.hu (T.P.); 2MTA-SZTE “Lendület” Fungal Pathogenicity Mechanisms Research Group, University of Szeged, Közép fasor 52, H-6726 Szeged, Hungary

**Keywords:** extracellular β-galactosidase, zygomycetes, activity screening, fermentation, transgalactosylation, transfructosylation, functional oligosaccharides, probiotics

## Abstract

β-Galactosidases of Mucoromycota are rarely studied, although this group of filamentous fungi is an excellent source of many industrial enzymes. In this study, 99 isolates from the genera *Lichtheimia*, *Mortierella*, *Mucor*, *Rhizomucor*, *Rhizopus* and *Umbelopsis*, were screened for their β-galactosidase activity using a chromogenic agar approach. Ten isolates from the best producers were selected, and the activity was further investigated in submerged (SmF) and solid-state (SSF) fermentation systems containing lactose and/or wheat bran substrates as enzyme production inducers. Wheat bran proved to be efficient for the enzyme production under both SmF and SSF conditions, giving maximum specific activity yields from 32 to 12,064 U/mg protein and from 783 to 22,720 U/mg protein, respectively. Oligosaccharide synthesis tests revealed the suitability of crude β-galactosidases from *Lichtheimia ramosa* Szeged Microbiological Collection (SZMC) 11360 and *Rhizomucor pusillus* SZMC 11025 to catalyze transgalactosylation reactions. In addition, the crude enzyme extracts had transfructosylation activity, resulting in the formation of fructo-oligosaccharide molecules in a sucrose-containing environment. The maximal oligosaccharide concentration varied between 0.0158 and 2.236 g/L depending on the crude enzyme and the initial material. Some oligosaccharide-enriched mixtures supported the growth of probiotics, indicating the potential of the studied enzyme extracts in future prebiotic synthesis processes.

## 1. Introduction

β-Galactosidases (β-D-galactoside galactohydrolase or lactase, EC 3.2.1.23) are important biocatalysts in many food and pharmaceutical applications, as they can catalyze both the hydrolysis and synthesis of glycosides affecting human health. For instance, β-galactosidases are frequently used to produce lactose hydrolyzed foods through the degradation of lactose to galactose and glucose monomers in the dairy industry [1]. These special food products are beneficial for people with low intestine β-galactosidase levels that causes the condition of lactose intolerance. The hydrolyzing activity of β-galactosidases is utilized in various other applications, such as prevention against lactose-related crystallization in frozen foods and condensed milk, or ethanol and sweet syrup production from whey lactose [2].

β-Galactosidases can synthesize galacto-oligosaccharide (GOS) and lactulose-derived GOS (OsLu) molecules through the enzymatic transgalactosylation of lactose and lactulose, respectively. The biosynthesis of these molecules is interesting, both in health and technological perspectives [3]. GOS and OsLu are non-digestible carbohydrates and serve as fermentable substances for beneficial gut microorganisms. They can stimulate the growth and/or activity of probiotics, including *Lactobacillus* and *Bifidobacterium* strains [4,5]. Practically, GOSs can be used in the development of prebiotic and bifidogenic food and beverage ingredients in the food technology as well as in the treatment of certain gastrointestinal disorders [6,7], which is also supported by their low caloric value and high stability at a wide range of pH and temperature [8]. Taken together, the discovery of new β-galactosidases from natural sources and/or by using metagenomic approaches, and characterization of their hydrolytic and synthetic capabilities are efficient ways to provide valuable catalysts for future glycobiology research [3].

Most commercial β-galactosidases are derived from bacteria, yeasts and filamentous fungi [9]. The production of the enzyme protein can be conducted using such organisms due to the possibility of mass cultivation, mild fermentation conditions as well as the easy elaboration of optimized production systems. In addition, many microorganisms, especially filamentous fungi, can grow well on low-cost media such as agro-industrial byproducts and utilize them to produce β-galactosidases in high amounts. The production and characterization of β-galactosidases have been extensively studied in fungi [2,9], and even the GOS synthesis has been evaluated in detail for many enzymes [6,10,11,12]. Mucoromycota fungi (i.e., members of the former Zygomycota), however, have been given less attention in this regard, although these microorganisms are good sources of many industrial enzymes, such as lipases, hydroxylases, phytases and a set of various carbohydrate cleaving biocatalysts [13]. To our knowledge, there are only some examinations dealing with extracellular β-galactosidases in this filamentous fungal group. These works mainly focused on enzyme activities from *Rhizomucor* isolates [14,15,16,17], but an effective producer *Mucor* sp. isolate has also been described by Silvério et al. [12].

Submerged fermentation (SmF) conditions are frequently used for β-galactosidase production by fungi. Excellent supports to achieve high enzyme yield in liquid conditions are lactose, skim milk and other materials that contain β-galactosyl linkage between some of their molecules. However, a number of studies have highlighted the relevance of the solid-state fermentation (SSF) approach on agrowaste substances, e.g., wheat bran and straw, rice bran and straw, sugarcane bagasse and pomegranate peel, for large scale β-galactosidase production in fungi [17,18,19,20].

In the present study, the β-galactosidase activity of Mucoromycota strains belonging to the order Mucorales and Mortierellales was evaluated and compared through a visually detectable chromogenic agar assay. The research included investigations on the enzyme production in both SmF and SSF conditions, as well as a comprehensive analysis on the oligosaccharide synthesis capacity of selected β-galactosidase-active crude extracts. Finally, the effect of oligosaccharide-enriched cocktails produced enzymatically on the growth of probiotic microorganisms was also investigated.

## 2. Materials and Methods

### 2.1. Microorganisms

The fungal and bacterial strains used in this study were obtained from the Szeged Microbiological Collection (Szeged Microbiological Collection (SZMC), Szeged, Hungary; http://www.wfcc.info/ccinfo/collection/by_id/987, accessed on 4 February 2021).

Fungal isolates of the genera *Lichtheimia*, *Mortierella*, *Mucor*, *Rhizomucor*, *Rhizopus* and *Umbelopsis* were subjected to the β-galactosidase activity screening test (Appendix A). Strains with the highest activity were selected and used for further enzyme production and activity studies. Before each experiment, the fungi were grown on malt extract medium (20% *v/v* malt extract, 50 mL/L; yeast extract, 5 g/L; glucose, 5 g/L; agar, 20 g/L) for 5–10 days at 20, 25 or 37 °C, depending on the culturing requirements of the applied strain. To prepare the spore suspension, a volume of 5 mL of distilled water was added to each slant. After vortexing, the sporangiospore number was set by serial dilution in distilled water followed by counting in a Bürker chamber under a light microscope.

The probiotic strains selected for growth-promoting activity assays were *Lactobacillus casei* SZMC 23430, *Lactobacillus acidophilus* LA-5 SZMC 23432, *Bifidobacterium animalis* subsp. *lactis* BB-12 SZMC 26956 and *Saccharomyces boulardii* CNCM I-745 SZMC 26957. Strains were maintained on de Man, Rogosa and Sharpe (MRS) agar (VWR, Debrecen, Hungary) at 37 °C. Before the assay, fresh cultures were prepared through cultivation in 30 mL of MRS broth (VWR) for 24 h at 37 °C under oxygen-depleted conditions generated by an Anaerocult C system (Merck, Budapest, Hungary) in a cultivation jar.

### 2.2. Chromogenic Agar Test for β-Galactosidase Activity Screening

A chromogenic method was used to screen the fungi for their β-galactosidase activity in Petri plate incubation. The growth medium contained 20% (*v/v*) malt extract (20 mL/L), lactose (20 g/L), peptone (1 g/L) and agar (20 g/L). After sterilization, the medium was supplemented with 0.5% (*v/v*) of 5-bromo-4-chloro-3-indolyl-β-D-galactopyranoside (X-gal; Thermo Fisher Scientific, Waltham, USA) solution (20 mg/mL in dimethyl-sulfoxide). A volume of 20 µL of spore suspension (10^6^ spores/mL) was added to each agar plate and the cultures were incubated for 10 days at temperatures depending on the culturing requirements of the tested fungus. Visual detection of the plates was performed daily, and the level of the β-galactosidase production was evaluated from the intensity of the blue color developed. The categories of no blue color (NC), light blue color (+), darker blue color (++), blue color (+++), dark blue color (++++) and deep dark blue color (+++++) were defined to evaluate the chromogenic test. Three biological replicates were performed with each tested strain. The qualitative chromogenic assay serves as a preliminary test to screen for enzyme production as reported in other similar studies [12,21,22,23,24].

### 2.3. β-Galactosidase Production in Submerged and Solid Cultures

For SmF tests, 20 µL of sporangiospore suspension (10^6^) was transferred to 100-mL Erlenmeyer flasks containing 30 mL of growth medium (in g/L: peptone 4, yeast extract 4, KH_2_PO_4_ 2, Na_2_HPO_4_*12 H_2_O 8, MgSO_4_*7 H_2_O 0.25) supplemented with lactose or lactose and wheat bran in a concentration of 20 g/L. Incubation was performed under continuous shaking (200 rpm) for 7 days at 20, 25 or 37 °C. A volume of 1 mL samples were taken immediately after the inoculation and on the 2nd, 4th, and 7th days of fermentation. Then, the samples were centrifuged at 16,200× *g* for 20 min and the supernatant diluted in sodium acetate buffer (50 mM, pH 6.0) was used to determine the β-galactosidase activity and the protein content.

The SSF was carried out in 250-mL Erlenmeyer flasks containing 5 g of wheat bran and 1 g of soy flour moistened with 5 mL of distilled water. After sterilization, the flasks were inoculated with 20 µL of sporangiospores (10^6^) and incubated for 7 days at 20, 25 or 37 °C. Sampling was scheduled as the same as to that applied in the liquid culture tests. Corresponding flasks were taken at predetermined intervals, and the fermented medium was extracted with 30 mL of sodium acetate buffer (50 mM, pH 6.0) by incubating the extract at 4 °C for 4 h. The extracts were then filtered through a gauze, and a 1 mL sample was taken from the filtrate and centrifuged at 16,200× *g* for 20 min. Then, the supernatant was stored at −20 °C until the β-galactosidase activity and total protein concentration assays. The samples were diluted in sodium acetate buffer (50 mM, pH 6.0) before the measurements. All fermentation tests were carried out in three independent experiments.

### 2.4. β-Galactosidase Activity Assay

β-Galactosidase activity was determined by incubating 180 μL of diluted samples at 50 °C for 30 min with 20 μL of 7 mM *o*-nitrophenyl-β-D-galactopyranoside (*o*NPG; Sigma–Aldrich, Munich, Germany) solution prepared in sodium acetate buffer (50 mM, pH 6.0). The reaction was stopped by adding 50 μL of 100 mM sodium carbonate, then the released *o*-nitrophenol was measured at 420 nm using a SPECTROstar Nano (BMG Labtech, Offenburg, Germany) microplate reader. A blank measurement was always performed before the incubation. The standard curve was set up from the absorbance data of the *o*-nitrophenol (Sigma–Aldrich) compound dissolved in sodium acetate buffer (50 mM, pH 6.0) in the concentration range of 20–200 µM. One unit of enzymatic activity was defined as the amount of enzyme that liberated 1 µM of *o*-nitrophenol per 1 min under the conditions of the assay.

### 2.5. Determination of the Protein Content

The total protein content in the culture filtrates and crude extracts was determined by using a Qubit Fluorometer (Invitrogen, Carlsbad, CA, USA) and the Quant-iT Protein Assay Kit (Invitrogen).

### 2.6. Partial Purification of β-Galactosidases

Seven-day-old cultures of wheat bran-based SSF were extracted by 30 mL of sodium acetate buffer (50 mM, pH 6.0), and after a 4-h incubation at 4 °C, the extracts were filtered through a gauze. To eliminate salts, low molecular weight sugars and other small molecules, a volume of 10 mL of each extract was filtrated through a Bio-Gel P-6 desalting cartridge (exclusion range 1 to 6 kDa; 50 mL; Bio-Rad, Hercules, CA, USA), which was equilibrated with 50 mM of sodium acetate buffer (pH 6.0). Elution was performed with the same buffer at a flow rate of 9 mL/min, and the fraction containing β-galactosidase activity was collected.

### 2.7. Gel Electrophoresis of Proteins

Sodium dodecyl sulfate-polyacrylamide gel electrophoresis (SDS-PAGE) was performed on 12% Mini-Protean TGX gel (Bio-Rad) using NuPAGE MES running buffer (Invitrogen) according to the manufacturer’s instructions. Protein bands were detected by staining the gels with 0.0025% Coomassie Brilliant Blue R-250.

### 2.8. Enzymatic Synthesis Assay

The synthesis of oligosaccharides was studied in four sets of reactions prepared in sodium acetate buffer (50 mM, pH 6.0) in a final volume of 2 mL. These reaction conditions contained (i) 15% (*w/v*) lactose, (ii) 10% (*w/v*) skim milk (48% *w/w* lactose content, Sigma–Aldrich), (iii) 10% (*w/v*) lactose + 10% (*w/v*) fructose or (iv) 90 mM *o*NPG + 10% (*w/v*) sucrose initial materials. A reaction mixture containing 20% (*w/v*) lactose and fructose was also prepared to examine the effect of the initial substrate concentration on product yield. After setting up the solutions, crude β-galactosidase corresponding to 2400 U activity was added, and the mixtures were incubated at 50 °C for 12 h under constant stirring (200 rpm). An enzyme-free sample of each reaction condition was used as the corresponding control. After incubation, the reaction mixtures were kept in boiling water for 5 min to stop the reaction. The samples were then cooled down to room temperature and stored at −20 °C until the analytical measurements and growth-promoting activity tests. All enzymatic synthesis assays were performed in three biological parallels. The oligosaccharide yield was calculated using the following equation:Oligosaccharide yield (%) = C_1_/C_0_ × 100(1)
where C_0_ and C_1_ are the initial concentration of the substrate (i.e., lactose, lactose content of skim milk or sucrose) and the concentration of oligosaccharides, respectively.

### 2.9. HPLC-MS/MS Analysis of Carbohydrates

The synthesized carbohydrates were analyzed by LC-MS/MS on a Nexera XR HPLC system (Shimadzu, Duisburg, Germany) coupled to a TSQ Quantum Access triple quadrupole mass spectrometer (Thermo Fisher Scientific) equipped with an H-ESI probe.

Liquid chromatographic separation was performed using a SeQuant ZIC-HILIC column (3.5 μm, 150 x 2.1 mm; Merck) equipped with a SeQuant ZIC-HILIC guard column (20 × 2.1 mm) thermostated at 25 °C. The mobile phase A consisted of 5 mM of ammonium acetate containing 0.1% formic acid, while acetonitrile containing 0.1% formic acid served as the mobile phase B. The gradient elution was performed as follows: 0 min, 80% B; 0.5 min, 80% B; 8.5 min, 40% B; 10.5 min, 40% B; 11 min, 80% B; 20 min, 80% B. The flow rate of the mobile phase was maintained at 200 µL/min and the injection volume was 5 µL.

The general MS conditions were set as follows: spray voltage, 4500 V; vaporizer temperature, 50 °C; sheath gas (nitrogen) pressure, 50 psi; auxiliary gas (nitrogen) flow, 10 arbitrary units; ion transfer capillary temperature, 200 °C; collision gas (argon) pressure, 1.5 mTorr. Electrospray ionization was operated at negative mode. Mass spectrometric detection of the carbohydrates was carried out in selected reaction monitoring (SRM) mode. SRM transitions were 226.0 > 180.3, 387.5 > 180.19, 549.2 > 180.2, and 711.2 > 383.5 for mono-, di-, tri- and tetrasaccharides, respectively. The acquired data were processed using Xcalibur™ version 2.2.1 and Trace Finder version Section 3.3 (Thermo Fisher Scientific). Carbohydrate standards (glucose, fructose, galactose, sucrose, lactose, lactulose, raffinose, 1-kestose and nystose; Sigma–Aldrich) were used to analyze individual compounds in the reaction mixtures.

### 2.10. Growth-Promoting Activity Assay

A liquid culture approach was used to investigate the effect of oligosaccharide-enriched samples on the growth of selected probiotics (see 2.1.). Half tubes containing 4 mL of 1% (*w/v*) skim milk solution, 900 µL of distilled water and 100 µL of reaction samples were inoculated with 100 µL of fresh probiotics suspension prepared in MRS broth. Control cultures consisted of 5 mL of 1% (*w/v*) skim milk solution and 100 µL of probiotics suspension (Control 1), or 4 mL of 1% (*w/v*) skim milk solution, 900 µL of distilled water, 100 µL of enzyme-free reaction sample and 100 µL of probiotics solution (Control 2). The culturing tubes were then incubated for 24 h at 37 °C under oxygen depleted conditions. Subsequently, a volume of 10 µL from the cultures were inoculated equidistantly onto MRS agar medium prepared in Petri dishes. After drying, the plates were placed into an anaerobic jar and incubated at 37 °C for 48 h. Colonies were counted and the colony forming unit (CFU) was calculated by using the following equation:CFU/mL = A × 10*^n^* × 100(2)
where A is the number of the probiotic microorganism colonies counted and *n* is the degree of dilution. Three biological replicates were performed with each sample tested.

### 2.11. Statistical Analysis

The results were calculated from at least two biological and three technical parallels and data were expressed as the means ± standard deviation. Significance was calculated with a one-way analysis of variance (ANOVA) followed by Tukey’s multiple comparison test in the GraphPad Prism 6.00 software (GraphPad Software Inc., San Diego, CA, USA). A *p* value of <0.05 was considered as statistically significant. Pearson’s correlation test in Microsoft 365 Excel was performed to investigate the association between the oligosaccharide concentration and the growth-promoting activity of samples.

## 3. Results

### 3.1. Screening of β-Galactosidase Activity

In this assay, the β-galactosidase activity of 99 fungal isolates from six Mucoromycota genera, namely *Lichtheimia* (18), *Mortierella* (16), *Mucor* (17), *Rhizomucor* (11), *Rhizopus* (20) and *Umbelopsis* (17), was screened. The cultivation of the strains on X-gal-containing medium was performed and the results were evaluated according to the intensity of blue color that developed on and around the colonies.

The enzyme production capacity varied within the tested fungi and there were notable differences in the activity between the strains from the same genus. Most strains with high activity were found among the *Lichtheimia*, *Rhizomucor* and *Umbelopsis* genera (Appendix A). Representatives of *Mortierella*, *Mucor* and *Rhizopus* were generally less active, however, the *Mortierella echinosphaera* SZMC 11251, *Mortierella globulifera* SZMC 11209, *Mucor plumbeus* SZMC 12023 and all tested *Rhizopus microsporus* strains can be highlighted as effective β-galactosidase producers from these groups (Appendix A). The activity data and origin of ten isolates identified as the best producers, namely *Lichtheimia ramosa* SZMC 11360, *Lichtheimia corymbifera* SZMC 11361, *Lichtheimia hyalospora* SZMC 11364, *Rhizomucor miehei* SZMC 11005, *R. miehei* SZMC 11014, *Rhizomucor pusillus* SZMC 11025, *Rhizopus microsporus* var. *oligosporus* SZMC 13619, *M. echinosphaera* SZMC 11251, *Umbelopsis longicollis* SZMC 11208 and *Umbelopsis ramanniana* var. *angulispora* SZMC 11234, and the temperature for their cultivation during the screening test were summarized in Table 1. A notable feature of most of these isolates was the high activity that appeared at the earlier stage of the incubation. After one-day cultivation, for instance, the *R. miehei* SZMC 11014 developed a visually well distinguishable blue color as compared to the other fungi. However, the activity of all the best producers reached a strong positive response if the later stage of incubation was considered (Table 1), and the blue color that developed turned to a deep dark at the 10th incubation day (Figure 1A–J).

### 3.2. β-Galactosidase Production

Strains that showed the best β-galactosidase activity in the screening tests (Table 1) were selected for fermentation assays. Both SmF and SSF conditions were carried out to investigate the enzyme production of the selected strains. Two approaches were used in the SmF experiments, in which the effect of the presence of lactose and wheat bran on extracellular β-galactosidase production yield was monitored. The SSF system consisted of wheat bran and soy flour dry matter moistened with distilled water.

Figure 2 depicts the best volumetric (U/mL medium or crude extract) and specific (U/mg protein) activity yields achieved during the fermentations. The weight of fungal biomass in wheat bran-contained systems could not be estimated due to separation difficulties of mycelia from the culture flasks. For comparison, specific activity data were considered and refers to the protein concentration in both the culture broths and the crude extracts.

Overall, the activity data varied between fungus-to-fungus and the β-galactosidase producing abilities differed according to the applied fermentation system. In lactose supplemented SmF medium, for instance, the *U. longicollis*, *L. hyalospora* and *R. pusillus* exhibited the highest volumetric activity with 48, 505 and 860 U/mL medium mean enzyme yields, respectively (Figure 2A). The *R. miehei* SZMC 11005 and *U. ramanniana* var. *angulispora* were less active (*p* < 0.05) with moderate volumetric activity yields of 14 and 20 U/mL medium, respectively. Even less activity was registered for the other fungi tested in this condition (Figure 2A). Concerning the specific activity, however, only the *L. hyalospora* and *R. pusillus* displayed outstanding (*p* < 0.05) β-galactosidase production (yields of 551 and 954 U/mg protein, respectively) among the tested fungi under this condition (Figure 2A). The *U. longicollis* exhibited a less specific activity (44 U/mg protein) that was not significantly (*p* > 0.05) different from the other tested fungi (Figure 2A).

The addition of wheat bran to the lactose-containing SmF medium greatly supported the β-galactosidase production yields. In general, both the volumetric and the specific activities increased by about 10–100 times as compared to the system containing lactose as the sole inducer. It is worth mentioning that a 1,000-fold increase was observed in the case of the *R. miehei* SZMC 11014 isolate (see Figure 2A,B). Interestingly, the activity of *L. hyalospora* in this SmF system reduced by about four times to that measured in SmF containing only lactose. Anyway, enzyme activity of the *Lichtheimia* and the *R. miehei* SZMC 11005 isolates remained moderate in this condition compared to the other tested fungi (*p* < 0.05). The β-galactosidase yield detected for these strains ranged between 50 and 303 U/mL medium, which indicates an efficient induction of enzyme production (Figure 2B). The highest activity yields obtained in this condition were 9,926 and 13,874 U/mL medium, which was exhibited by the *R. miehei* SZMC 11014 and *R. pusillus* isolates, respectively (Figure 2B). Interestingly, enzyme production of the former strain was among the lowest in the system containing lactose as the sole inducer. *R. pusillus*, however, showed high enzyme yield in that fermentation experiment (Figure 2A). The tested *R. microsporus* var. *oligosporus*, *M. echinosphaera* and the two *Umbelopsis* isolates did not show any significant difference in their volumetric activities (*p* > 0.05). In contrast, *R. microsporus* var. *oligosporus* and *M. echinosphaera* resulted in significantly higher specific activity compared to the *Umbelopsis* isolates (*p* < 0.05). The maximum β-galactosidase activities of these strains were found to be varying between 1109 and 1387 U/mL medium (from 675 to 1436 U/mg protein) (Figure 2B).

For most fungi tested, cultivation in the wheat bran-based SSF system further improved the β-galactosidase yields obtained during the two SmFs. Namely, the SSF resulted in a more than 50-fold and 100-fold increase in the volumetric and specific activities, respectively, for *L. ramosa*, *L. hyalospora* and *R. miehei* SZMC 11005 isolates compared to those measured in wheat bran-supplemented SmF (see Figure 2B,C). In this context, elevated specific activity yields were also registered for the other tested strains, except for the *M. echinosphaera* and *R. miehei* SZMC 11014, but to a lesser extent than those observed for the above fungi. For *U. longicollis* and *U. ramanniana* var. *angulispora*, the volumetric activity was slightly higher (1202 and 1248 U/mL, respectively) in wheat bran-based SmF to that achieved in SSF (1109 and 806 U/mL, respectively). Anyway, in SSF, the overall enzyme production of *Lichtheimia*, *Rhizomucor* and *Rhizopus* fungi was superior to those of the *Umbelopsis* and *Mortierella* isolates. As in the two SmF tests, *R. pusillus* exhibited the best activity during SSF, reaching maximal volumetric and specific activity yields of 19,448 U/mL crude extract and 22,720 U/mg protein, respectively (Figure 2C). Considering the amount of wheat bran substrate subjected to fermentation, the maximal β-galactosidase activities ranged from 3308 to 116,690 U/g substrate (Figure 3).

### 3.3. Synthesis of Oligosaccharides

The oligosaccharide synthesis capacity of β-galactosidase-active cocktails from *L. ramosa* and *R. pusillus* obtained on wheat bran-based SSF was also studied. Four sets of conditions were applied in a parallel experiment, each containing different compounds as the glycosyl donor and/or acceptor. Namely, lactose, skim milk, lactose–fructose and *o*NPG–sucrose-based mixtures were prepared and incubated with or without the corresponding enzyme under a standard condition (50 °C, 12 h). Before the reactions were compiled, the β-galactosidase ferments were purified to eliminate the saccharides and other compounds, which could interfere with the product detection. Protein cocktails with β-galactosidase activity of 6233 ± 139 U for *L. ramosa* and 11,954 ± 113 U for *R. pusillus* were successfully prepared through the gel filtration approach used in the study. The protein composition of the cocktails was also analyzed. SDS-PAGE detected protein bands with molecular weights ranging from 15 to 90 kDa and from 27 to 90 kDa in *L. ramosa* and *R. pusillus* samples, respectively (Appendix A). Crude β-galactosidases corresponded to 2400 U activity were introduced to each reaction mixture, and after the incubation, the formation of oligosaccharides was examined via the HPLC MS/MS technique.

Analytical measurements detected oligosaccharides under all reaction conditions, which proved the synthetic activity of the crude *L. ramosa* and *R. pusillus* β-galactosidases. The amount of tri- and tetrasaccharides obtained and the oligosaccharide yield are summarized in Table 2. It shows that the condition based on *o*NPG-sucrose initial material resulted in the highest oligosaccharide concentrations and the overall synthetic capacity of the *R. pusillus* crude β-galactosidase were superior to those measured for the *L. ramosa* enzyme.

Oligosaccharide synthesis reactions on lactose and skim milk were performed to estimate transgalactosidase activity. In this type of reaction, tri- and tetrasaccharide GOS products formed by the transgalactosylation of lactose were identified (Table 2). After 12 h incubation at 50 °C, the GOS contents on skim milk (with a lactose concentration of 48 g/L) by *L. ramosa* and *R. pusillus* enzymes achieved 15.8 and 22.7 mg/L, respectively. When 150 g/L of lactose was the initial material and the incubation parameters were the same as for skim milk, the GOS concentrations reached maximums at 164 and 257.6 mg/L for *L. ramosa* and *R. pusillus* β-galactosidases, respectively.

The synthesis of lactulose, which is the basis of OsLu compounds, was examined in reaction mixtures which contained lactose and fructose as initial sugars. This system contained fructose as the acceptor of galactose to produce lactulose disaccharide. However, no lactulose was identified under the applied reaction conditions. Tests were carried out with 20% (*w/v*) initial lactose and fructose concentration as well, but no lactulose was detected even in this system. However, under this condition, about two- and 1.2-fold increases in the trisaccharide concentration were achieved for *L. ramosa* and *R. pusillus* crude β-galactosidases, respectively, as compared to the sample containing 10% (*w/v*) of initial substrate (Figure 4). Interestingly, the elevated initial sugar concentration did not affect the amount of tetrasaccharide obtained. The formation of fructo-oligosaccharide (FOS) components, i.e., 1-kestose and nystose, was also monitored. These molecules could not be detected even at initial sugar concentrations of 10% and 20% (*w/v*), and therefore, the resulted oligosaccharides might be the transgalactosylation products of the lactose.

Intermolecular transgalactosylation reactions were also monitored on sucrose as the acceptor and *o*NPG as the galactose donor molecule. This combination of initial materials also resulted in tri- and tetrasaccharides (Table 2). However, FOS compounds were detected, suggesting the simultaneous action of various enzymatic activities in the crude enzyme cocktails. Namely, the reaction mixtures prepared by enzymatic extracts from *L. ramosa* and *R. pusillus* contained the FOS molecules 1-kestose (6.5 ± 0.3 and 100.9 ± 1.1 mg/L, respectively) and nystose (28.2 ± 1.1 and 114.2 ± 1.7 mg/L, respectively). In addition, a trisaccharide product was also obtained with a mass-to-charge ratio (*m/z*) of 503.3 in this reaction. Concentrations of this compound were 90.3 ± 5.4 and 199.3 ± 12.7 mg/L for *L. ramosa* and *R. pusillus* crude β-galactosidases, respectively.

### 3.4. Growth-promoting Activity of Synthesized Oligosaccharides

The effect of oligosaccharide-enriched mixtures produced with both the *L. ramosa* and the *R. pusillus* crude β-galactosidases on the growth of selected probiotic microorganisms was investigated. In this study, the CFU of *L. casei*, *L. acidophilus* LA-5, *B. animalis* subsp. *lactis* BB-12 and *S. boulardii* CNCM I-745 was monitored in the presence of reaction samples prepared with initial materials of lactose–fructose, skim milk or lactose. The reaction mixture containing *o*NPG as the glycosyl donor was excluded from the experiment due to the potential growth inhibitory effect of the liberating *o*-nitrophenol. The colony counts were compared to controls, that is, cultures incubated without the addition of reaction samples were considered as Control 1, while Control 2 contained the enzyme-free sample of each reaction mixture. Controls were prepared separately to each cultivation system. After incubation for 24 h, there was no statistically significant difference between the colony counts of the Control 1 and Control 2 (Figure 5A–D), which means that the enzyme-free reaction mixture alone, regardless of the initial sugars, did not affect the bacterial growth.

For most tests, the oligosaccharide-enriched samples stimulated the bacterial growth as compared to the corresponding controls. In *L. casei*, for instance, the colony count analysis showed a 1.02 to 1.03 times increase in CFU after incubation with enzyme treated samples (*p* < 0.05) (Figure 5A). The highest growth-promoting effect to *L. casei* was observed for oligosaccharide-containing samples obtained on lactose–fructose initial sugars. The overall colony count in these treated cultures was 8.73 log CFU/mL, while it was about 8.48 log CFU/mL for the controls (Figure 5A). In the case of *L. acidophilus*, the highest increase in the colony count was 1.08-fold (from 6.28 to 6.77 log CFU/mL) compared to the control (*p* < 0.05) and it was registered for enzyme-treated samples containing skim milk as the initial sugar source (Figure 5B). Lactose–fructose systems treated by *L. ramosa* or *R. pusillus* enzymes, as well as the *R. pusillus*-supplemented lactose-based solution significantly promoted the *L. acidophilus* growth (*p* < 0.05). Additionally, all enzyme-containing samples significantly (*p* < 0.05) increased the count of *B. animalis* subsp. *lactis* colonies compared to the controls (Figure 5C). Concerning *S. boulardii*, the addition of a GOS-enriched mixture obtained on skim milk did not affect the colony count compared to the control samples (Figure 5D). When the enzyme treated samples containing lactose or lactose–fructose initial sugars were applied as additives, however, an increase in the colony count was observed. Among the reaction mixtures tested, the sample prepared on lactose–fructose initial sugars with *R. pusillus* crude β-galactosidase exhibited the highest growth-supporting activity to *S. boulardii* (from 6.57 to 6.91 log CFU/mL; *p* < 0.05) (Figure 5D).

Further analysis of the colony count data as correlated with the trisaccharide and tetrasaccharide concentrations obtained in each saccharide synthesis assay (see 3.3.) revealed a correlation between the growth-supporting activity of reaction mixtures and the amount of these oligosaccharides after the treatment with the crude β-galactosidase cocktails. This was strongly positive (*r* > 0.900) in *L. acidophilus* and *B. animalis* subsp. *lactis* for both tri- and tetrasaccharide contents found in all reaction samples (Table 3). Similar association (*r* > 0.900) was observed by *L. casei* and *S. boulardii* for both tri- and tetrasaccharide amounts obtained on lactose, and by *S. boulardii* for trisaccharide contents obtained on skim milk and lactose–fructose initial materials. Positive (0.800 ˂ *r* ˂ 0.900) and moderate (0.600 ˂ *r* ˂ 0.800) relationships of some studied variables were identified for *S. boulardii* and *L. casei* probiotics, respectively (Table 3).

## 4. Discussion

β-Galactosidases are important biocatalysts in the food industry where both the hydrolytic and the synthetic activities can be utilized. Although the production and the activity of β-galactosidases are well characterized in many microorganisms [9], a comprehensive analysis for Mucoromycota has not yet been performed. The current study demonstrates β-galactosidase producers among Mucorales and Mortierellales, inductive conditions for high-yield enzyme production, as well as the oligosaccharide synthesis capacity of selected Mucoromycota biocatalysts on various initial materials. In addition, the growth-promoting activity of oligosaccharide-enriched cocktails towards probiotics was also evaluated.

In screening assays, β-galactosidase production of 99 isolates from the genera *Rhizomucor*, *Rhizopus*, *Lictheimia*, *Umbelopsis*, *Mortierella* and *Mucor* was investigated through a chromogenic agar test. The results showed hydrolysis of X-gal by 66 strains, most of which belong to the *Rhizomucor*, *Lichtheimia* and *Umbelopsis* groups. There were five *R. microsporus* strains with notable activity in the *Rhizopus* group, and one isolate of *M. echinosphaera* and *M. globulifera* with promising β-galactosidase production were identified among *Mortierella*. It is worth mentioning that the same *M. echinosphaera* was found to be an excellent producer of a lipase with transesterification activity in our previous research [25]. The combined application of β-galactosidases and lipases can result in sugar–lactate copolymer biomaterials under appropriate conditions [26] and the possibility to utilize the two biocatalysts from a same organism is a particularly ecofriendly way. Within the *Mucor* group, *M. circinelloides* and *M. plumbeus* isolates presented the highest activity level. Some β-galactosidase producers have already been reported from the genera *Rhizopus*, *Rhizomucor* and *Mucor* [12,27,28], however, as far as we know, there were no data concerning *Lichtheimia*, *Mortierella* and *Umbelopsis* species. In filamentous fungi, the enzyme production of *Aspergillus*, *Penicillium* and *Trichoderma* species was studied in detail, and several producers have been identified during recent research studies [12,29,30,31,32]. Commercial β-galactosidases from filamentous fungal sources, especially from *Aspergillus*, were also previously prepared [33].

The yield of microbial β-galactosidase production can be increased by using culture conditions supplemented with different inductors. For instance, there were applications of skim milk, lactose, whey, and many agro-industrial substrates as additives to improve the enzyme yield in fungi [19,29,30,34]. In our study, the extractable β-galactosidase activity of ten producers selected following the screen tests (Table 1) was further analyzed in SmF and SSF systems containing lactose and/or wheat bran substrates as inducers for the enzyme production. Differences in β-galactosidase activities were more detectable in this assay than on X-gal medium. In all fermentation systems, *R. pusillus* demonstrated the highest volumetric and specific activities, but there were noticeable activity yields for *Lichtheimia* in SSF (from 3088 to 5793 U/mg protein) and for *R. microsporus* var. *oligosporus*, *M. echinosphaera* and the two *Umbelopsis* isolates tested (from 675 to 1436 U/mg protein) in wheat bran-supplemented SmF conditions. Additionally, both *R. miehei* strains proved to be good producers in the SSF system (about 7800 U/mg protein). Comparing the results of wheat bran-based fermentations, most of the maximal enzyme yields were achieved under the solid culture condition. However, *M. echinosphaera* and *R. miehei* SZMC 11014 presented slightly more activities in the wheat bran-based SmF than in SSF. Moreover, the SmF resulted in more volumetric activity yields for *Umbelopsis* than SSF. The different β-galactosidase activities measured in the SSF and SmF environment can be attributed to several reasons. For instance, the enzyme may be trapped in the cell wall under the SmF condition, while it is secreted to the medium in SSF systems, as it can be observed for other fungal hydrolases [35]. Furthermore, the thermostability of the enzyme can also be different depending on the producing environment applied [36]. It is also well known that culturing in SSF is close to natural growth conditions for filamentous fungi, in which the expression of hydrolytic enzymes is generally more supported [37].

Wheat bran is an inductive substrate for the growth and industrial enzyme production of filamentous fungi both in SSF and SmF conditions [38], which may be responsible for the higher β-galactosidase activity measured in wheat bran-containing media compared to lactose-based systems. The property of wheat bran to enhance the β-galactosidase production yield has been documented for other filamentous fungi. For instance, optimized yields were reached for *Aspergillus tubingensis* (15,936 U/g substrate) [18], *Penicillium canescens* (5292 U/g substrate) [39], and *Trichoderma* sp. (2.67 U/g substrate) [31] in SSF systems, and for *Trichoderma reesei* (25.43 U/mL) [40], *Aspergillus niger* (from 0.279 to 0.764 U/mL) [41] and *Aspergillus flavus* (843.75 U/mg protein) [42] in SmF systems.

Table 4 summarizes Mucoromycota fungi studied so far in terms of their β-galactosidase production. As seen, lactose was a commonly studied substrate for the production and the yields reported were generally comparable to those reached in our experiments (Figure 2A). The enzyme activity of *Rhizomucor* were tested in wheat bran-based SSF (Table 4). However, as we know, there were no SmF studies so far that used lactose and wheat bran together as a β-galactosidase inducer. Collectively, it can be concluded that wheat bran is an ideal substrate to induce the β-galactosidase production in Mucoromycota. The volumetric and specific activity data obtained in our study (Figure 2B,C) are significant compared to those of other Mucoromycota β-galactosidases presented previously (Table 4).

It is worth mentioning that the incubation temperature (50 °C) used in β-galactosidase activity assays may not have been optimal for all enzyme activities tested after the fermentation because the tested fungi cultured could have produce β-galactosidases at different temperature optimum. Nevertheless, it can be emphasized that most of the β-galactosidase activities showed by *M. echinosphaera*, *U. longicollis* and *U. ramanniana* var. *angulispora* (growth at 20 or 25 °C) were comparable to the activity data measured in other tested fungi (growth at 37 °C) (see Figure 2).

The power of β-galactosidases to catalyze enzymatic synthesis reactions was examined through the formation of oligosaccharides on different substrates used as glycosyl donors and/or acceptors in mixture. Considering that *L. ramosa* and *R. pusillus* exhibited a high β-galactosidase yield and activity during the SSF and the fact that these fungi are important producers of industrial biocatalysts [13,44,45,46], their enzymes were selected for these experiments. The β-galactosidases obtained in SSF were partially purified, and the crude enzymes were introduced to lactose, skim milk, lactose–fructose, or *o*NPG–sucrose containing reaction solutions. It is worth mentioning that protein composition analysis revealed a complex protein pattern for both the *L. ramosa* and the *R. pusillus* crude enzyme samples. Identification, purification, and characterization of their β-galactosidase enzymes are the subject of our future investigations.

The oligosaccharide synthesis capability of Mucoromycota β-galactosidases is a highly unexplored area. The literature data are available only for a *Mucor* sp. enzyme, in which authors reported 2 g/L and 2.6 g/L GOS concentrations after 20–30-h incubation with mixtures of lactose–fructose and lactose–sucrose, respectively [12]. In the present study, the tested enzymes also exhibited transgalactosylation activity and the maximal oligosaccharide concentrations ranged between 0.0158 and 2.236 g/L under non-optimized reaction conditions with 12-h incubation time.

GOS production was assessed on skim milk and lactose as initial materials. At the end of the incubation, oligosaccharides were formed on both substrates, indicating transgalactosylation action. Trisaccharide was the main product, similar to the findings of Rodriguez-Colinas et al. [47] who used skim milk and lactose initial substrates and a *Bacillus circulans* β-galactosidase (Biolactase) catalyst. The moderate oligosaccharide yield achieved on skim milk can be due to the lactose concentration present. However, skim milk may also contain components (e.g., cations) that can affect the activity of β-galactosidase during the reaction [48]. Anyway, oligosaccharides also appeared in skim milk fermentation, showing the potential of *L. ramosa* and *R. pusillus* β-galactosidases to be applicable in the dairy industry to produce GOS-enriched milk products.

Lactulose and GOS synthesis were studied when the crude *L. ramosa* and *R. pusillus* β-galactosidases were added to a mixture of lactose–fructose. Although oligosaccharide products were observed in this mixture, no lactulose was detected during the reaction at both 10% and 20% (*w/v*) initial lactose–fructose concentrations. It is important to note, however, that the approaches dealing with high-yield lactulose synthesis apply even higher initial lactose and fructose concentrations than those used in our experiments [49,50,51,52]. Practically, in our tests, an increase in the initial substrate concentration caused an increase in the trisaccharide amounts detected after the incubation. These molecules, and the tetrasaccharides, were formed by the transgalatosylation of the lactose. In addition, despite the fact that the *R. pusillus* enzyme resulted in more oligosaccharides than the *L. ramosa* β-galactosidase in all reactions performed, the dependence from the initial substrate concentration has markedly been observed for the latter enzyme here. Correlation between the initial lactose content and the resulting GOS yield has been described for many β-galactosidases. This relationship, however, can be varied from enzyme-to-enzyme and affected by the reaction parameters (e.g., temperature) [6,8].

Oligosaccharide formation was also observed when the crude enzymes were incubated in a mixture of *o*NPG and sucrose as the initial material. In this reaction, FOS components, i.e., 1-kestose (C18) and nystose (C24), were detected. In the work of Silvério et al. [12] performed on lactose–sucrose sugars, although that was also an unoptimized experiment, FOS synthesis by the *Mucor* sp. enzyme was not observed, but was described for most of the other fungal catalysts tested (e.g., for *Aspergillus*, *Penicillium* and *Trametes* enzymes). In most cases, 1-kestose has been identified as the main constituent of FOS. Kestose is also an important prebiotic [53] and can be synthesized by the transfructosylation activity of β-fructofuranosidase (FFase) or fructosyltransferase (FTase) enzymes [54,55]. Accordingly, the crude β-galactosidase cocktails purified in our study may have the transfructosylation activity that was responsible for the presence of FOSs during reactions with the mixture of *o*NPG–sucrose. In addition, because nystose is a common indicator sugar of transfructosylation activity [54,56,57], the presence of this compound in the *o*NPG–sucrose system further strengthens the occurrence of the reaction. Nevertheless, our results highlighted promising transfructosylation action of both the *L. ramosa* and the *R. pusillus* enzyme cocktails, that, although we have not yet investigated such enzyme activities, may be attributed to FFase and/or FTase enzymes present. Noticeable FFase or FTase activities have already been reported in zygomycetes, including strains of *Rhizopus stolonifer* [58], *Rhizopus delemar*, *Amylomyces rouxii* [59], *R. microsporus* [60] and *M. circinelloides* [61].

The transfructosylation activity may allow the synthesis of additional FOS compounds, such as lactosucrose (4^G^-β-D-galactosylsucrose, [62]) depending on the initial sugars. Moreover, certain β-galactosidases can also synthesize the lactosucrose [63,64,65]. In our study, analytical assays showed a trisaccharide compound with the same retention time and mass-to-charge ratio (*m/z* 503.3) of the raffinose in the *o*NPG–sucrose containing reaction. It is very likely, although a more detailed analysis of the product is needed, that the resulted saccharide is a lactosucrose, or probably an isoraffinose (6^G^-β-D-galactosylsucrose), or a mixture of these two compounds. These molecules, however, could not be separated by the HPLC-MS technique used in this study. With this context, it is worth mentioning that certain studies suggested the complete inability of fungal β-galactosidases to synthesize lactosucrose by transgalactosylation [12,66]. For isoraffinose synthesis via transgalactosylation action, Suyama et al. [67] reported an evidence in the *o*NPG–sucrose system using β-galactosidase from *Escherichia coli*. Finally, in comparison with lactose–sucrose-based systems commonly used for lactosucrose synthesis, glucose can be released only from the sucrose in such *o*NPG–sucrose approaches used in our study. Thereby, the lower glucose amount in the mixture can reduce its inhibitory effect on carbohydrase activities responsible for FOS synthesis.

The GOS and FOS molecules developed enzymatically can be used as prebiotic fibers in both adults and infants, promoting the activity of beneficial gut microorganisms [5,68]. In further experiments, we examined the growth-promoting potential of the prepared oligosaccharide-enriched mixtures, i.e., those containing lactose–fructose, skim milk or lactose as initial sugars, towards probiotics commonly used as additives in products. The tested *L. casei*, *L. acidophilus*, *B. animalis* subsp. *lactis* and *S. boulardii* probiotic microorganisms possess many properties beneficial for human health [69,70,71], including the action against pathogens [72,73,74]. The probiotic microorganisms responded to the oligosaccharide-enriched mixtures differently, however, in most cases, considerable growth-promoting effect was identified. In this way, prebiotic oligosaccharides may have formed in the reaction systems developed. On the other hand, the activity towards probiotics (Figure 5) was not altered according to the oligosaccharide content (Table 2) of the different reaction systems. Namely, the growth-supporting action of the skim milk-based system was comparable to those documented for lactose–fructose and lactose systems, except for the *S. boulardii* yeast. Skim milk proved to be a good support for the enzymatic (Maxilact β-galactosidase cocktail) production of oligosaccharides with prebiotic activity in the study of Oh et al. [75]. Comparing the effect of the samples obtained with the *R. pusillus* and *L. ramosa* crude β-galactosidases, mixtures prepared with the former biocatalyst generally resulted in a higher colony count than with the latter one. In line with this, correlation analysis revealed the association between the growth-promoting activity and the oligosaccharide content of a given sample. The higher the oligosaccharide content in a reaction sample after the enzyme treatment, the better the growth of probiotic microorganisms investigated in our study. This assumes that the oligosaccharides formed were responsible for the growth-promoting activity of the samples. However, prebiotic activity of the resulting oligosaccharides has not yet been proven by the above examinations. Therefore, our future investigations aim for the purification of the oligosaccharides formed and their use as sole carbohydrate source in fermentations [76].

## 5. Conclusions

Although microbial β-galactosidases are key catalysts in the bioprocess technology, there is a knowledge gap in terms of the enzyme activity and production in Mucoromycota. In this work, first, β-galactosidase-producing isolates from the Mucoromycota group were screened according to their X-gal hydrolyzing capacity. The results revealed the highest activities for strains belonging to the genera *Rhizomucor*, *Lichtheimia* and *Umbelopsis*. Next, we evaluated the β-galactosidase production of ten selected producers under inductive SmF and SSF conditions containing lactose and/or wheat bran as the growth substrate. We confirmed wheat bran as an excellent support for β-galactosidase production both in SmF and SSF systems. It is worthy to note that any of the fermentation conditions used in this study were not an optimized environment. The optimization of the culturing environment can further improve production yields, that is, the subject of future experiments. According to the enzymatic synthesis test, the *L. ramosa* and *R. pusillus* crude β-galactosidases investigated can synthesize oligosaccharides from lactose, skim milk, lactose–fructose or *o*NPG–sucrose initial substrates. Moreover, through FOS compounds formed on *o*NPG–sucrose initial material, we found that the *L. ramosa* and *R. pusillus* could also produce enzyme with transfructosylation activity. Finally, we confirmed the growth-promoting effect towards commercial probiotics for the oligosaccharide-enriched mixtures obtained in lactose, skim milk and lactose–fructose based conditions. In conclusion, our results provide useful data about the β-galactosidase producing capacity of Mucoromycota, including the groups of *Lichtheimia*, *Mortierella* and *Umbelopsis* that have not yet been investigated. The isolates with high hydrolytic activity can be reliable sources of β-galactosidases utilizable in the food industry. Additionally, the synthetic activities identified may contribute to the production of functional oligosaccharide mixtures with a high prebiotic index on appropriate initial sugars and conditions. These sugar cocktails then can be used as additives enhancing the health benefits of products.

## Figures and Tables

**Figure 1 jof-07-00229-f001:**
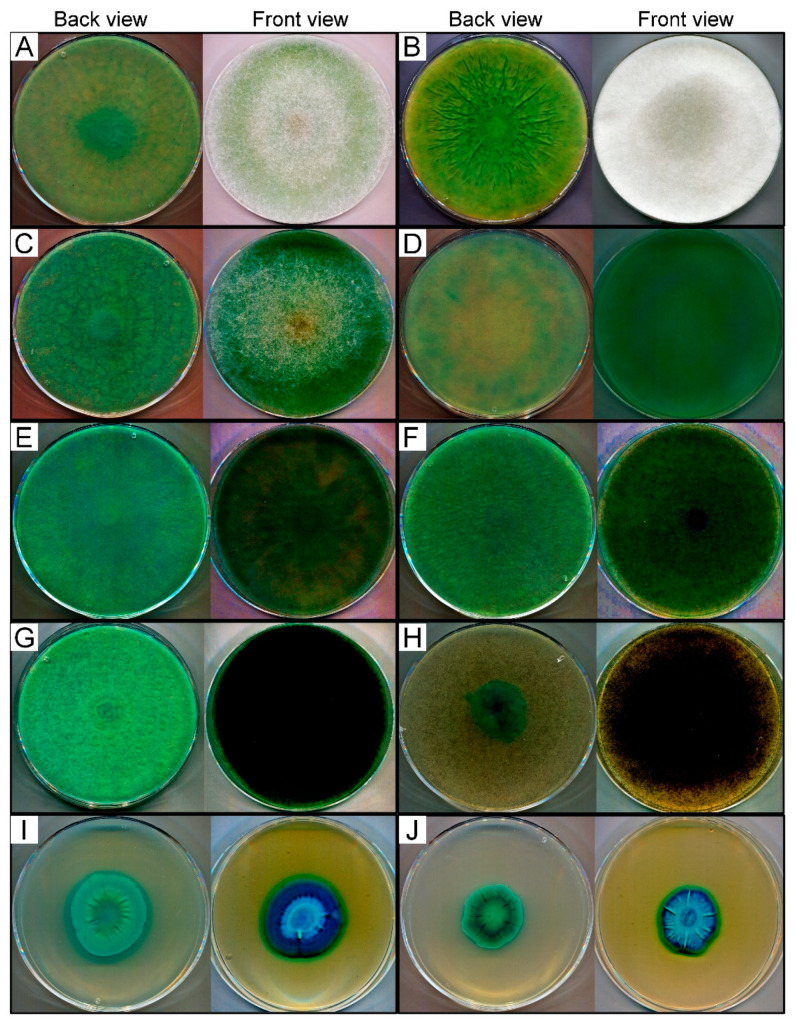
Colonies of *L. corymbifera* SZMC 11361 (**A**), *L. hyalospora* SZMC 11364 (**B**), *L. ramosa* SZMC 11360 (**C**), *M. echinosphaera* SZMC 11251 (**D**), *R. miehei* SZMC 11005 (**E**), *R. miehei* SZMC 11014 (**F**), *R. pusillus* SZMC 11025 (**G**), *R. microsporus* var. *oligosporus* SZMC 13619 (**H**), *U. longicollis* SZMC 11208 (**I**) and *U. ramanniana* var. *angulispora* SZMC 11234 (**J**) Mucoromycota strains grown on X-gal-containing medium. Experiments were performed on malt extract medium (20% *v/v* malt extract, 20 mL/L; lactose, 20 g/L; peptone, 1 g/L; agar, 20 g/L) containing 0.5% (*v/v*) X-gal solution. Data were recorded at the 10th day of cultivation.

**Figure 2 jof-07-00229-f002:**
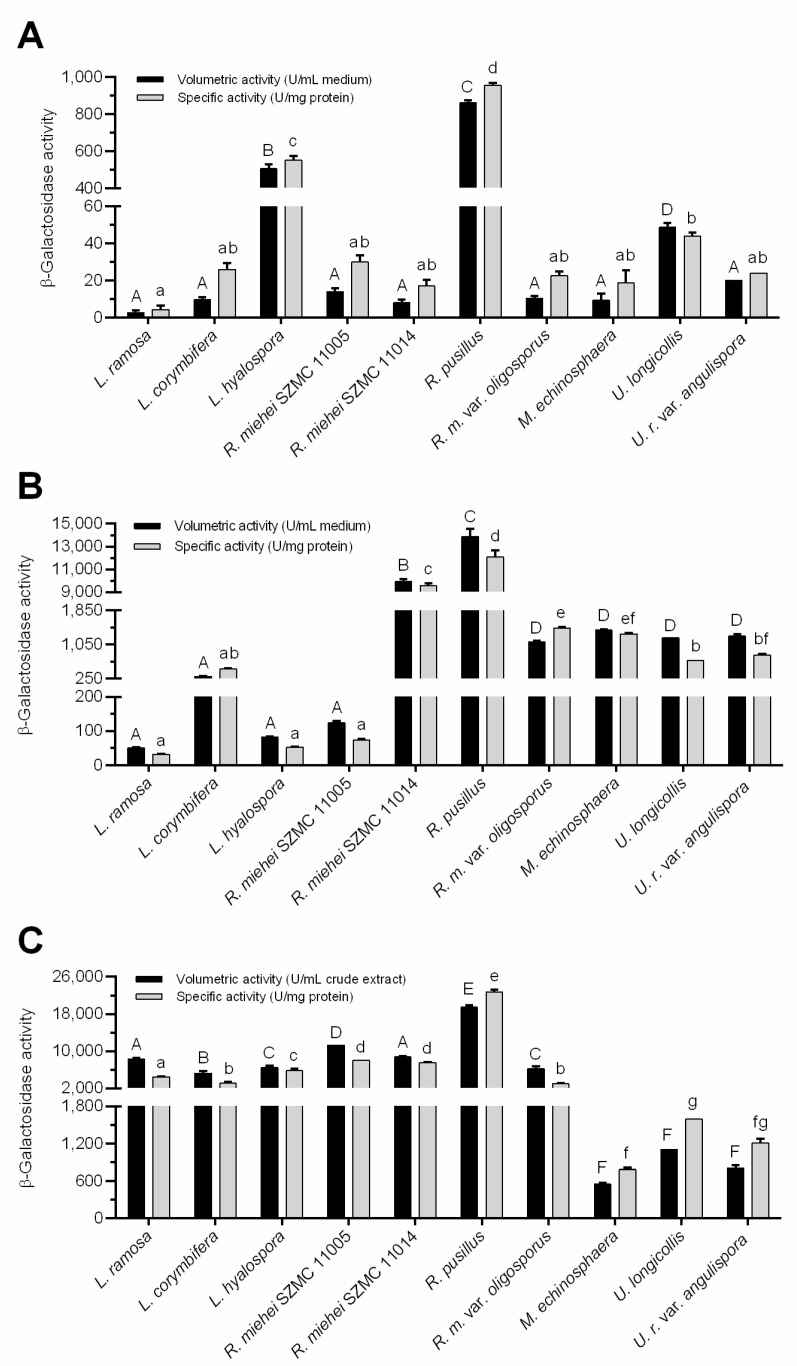
Comparative evaluation of best β-galactosidase yields (volumetric and specific activities) achieved with β-galactosidase-producing Mucoromycota fungi under different fermentation conditions. (**A**) Submerged fermentation (SmF) with medium containing 20 g/L lactose. Presented activities were measured on the 2nd day of incubation by *L. ramosa*, *L. corymbifera*, *R. miehei*, *R. microsporus* var. *oligosporus* and *U. ramanniana* var. *angulispora*, 4th day by *R. pusillus* and 7th day by *L. hyalospora*, *M. echinosphaera* and *U. longicollis*. (**B**) SmF with medium containing 20 g/L lactose and wheat bran. Presented activities were recorded on the 7th day except for *R. microsporus* var. *oligosporus* (2nd day) and *L. corymbifera*, *L. hyalospora*, *M. echinosphaera* and *U. longicollis* (4th day). (**C**) Wheat bran-based solid-state fermentation (SSF). Presented activities were determined on the 7th day of fermentation except for *M. echinosphaera*, *U. longicollis* and *U. ramanniana* var. *angulispora* (4th day). The cultivation temperature was 20 °C for *M. echinosphaera*, 25 °C for *U. longicollis* and *U. ramanniana* var. *angulispora*, and 37 °C for *L. ramosa*, *L. corymbifera*, *L. hyalospora*, *R. miehei*, *R. pusillus* and *R. microsporus* var. *oligosporus*. Values are averages computed from three replicates; error bars represent standard deviation. The different uppercase letters (within volumetric activity) or lowercase letters (within specific activity) above the columns indicate significant differences according to one-way ANOVA followed by Tukey’s multiple comparison test (*p* < 0.05).

**Figure 3 jof-07-00229-f003:**
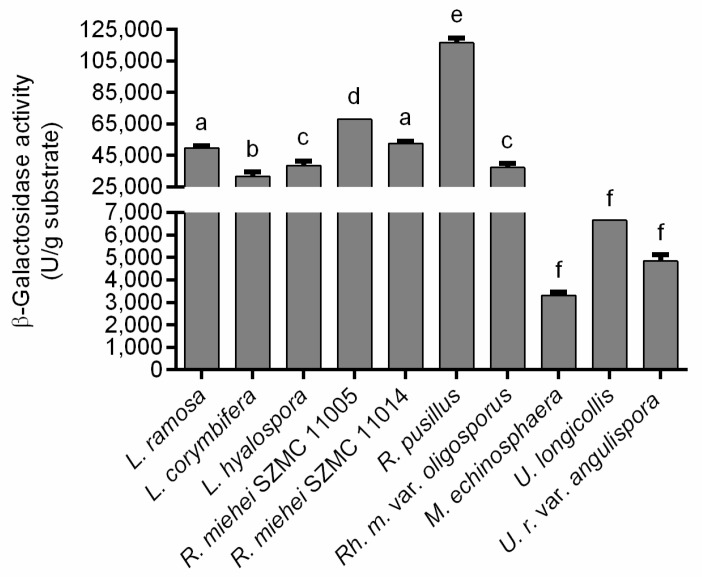
β-Galactosidase activity (in U/g substrate) of Mucoromycota in wheat bran-based SSF. Presented activities were determined on the 7th day of fermentation except for *M. echinosphaera*, *U. longicollis* and *U. ramanniana* var. *angulispora* (4th day). The cultivation temperature was 20 °C for *M. echinosphaera*, 25 °C for *U. longicollis* and *U. ramanniana* var. *angulispora*, and 37 °C for *L. ramosa*, *L. corymbifera*, *L. hyalospora*, *R. miehei*, *R. pusillus* and *R. microsporus* var. *oligosporus*. Values are averages computed from three replicates; error bars represent standard deviation. The different letters above the columns indicate significant differences according to one-way ANOVA followed by Tukey’s multiple comparison test (*p* < 0.05).

**Figure 4 jof-07-00229-f004:**
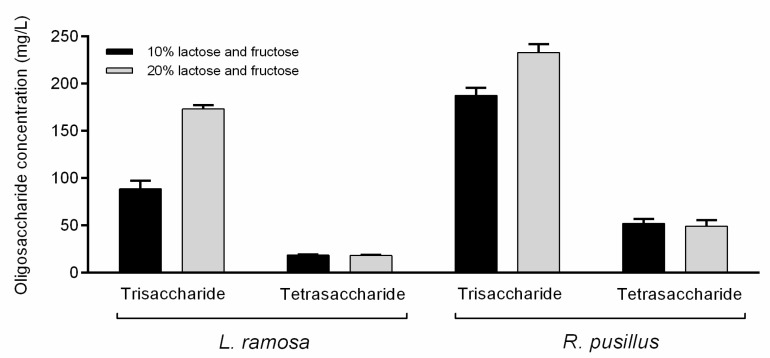
Comparison of oligosaccharide concentrations obtained by *L. ramosa* and *R. pusillus* crude β-galactosidases at 10% and 20% (*w/v*) initial sugar concentrations. The initial sugars were lactose and fructose in mixture. Reactions were performed in a volume of 2 mL at 50 °C for 12 h. Values are averages computed from three replicates; error bars represent standard deviation.

**Figure 5 jof-07-00229-f005:**
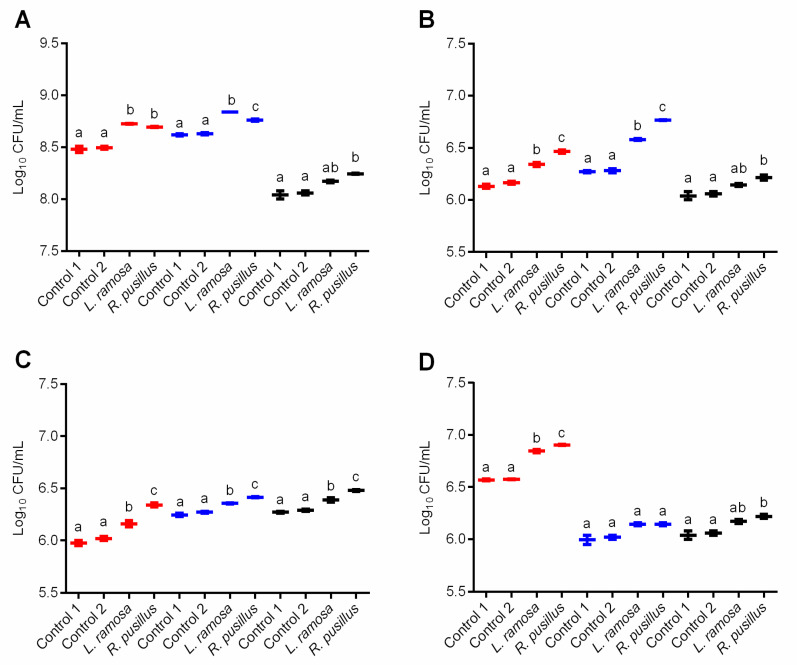
Effect of oligosaccharide-enriched solutions on the growth of *L. casei* (**A**), *L. acidophilus* LA-5 (**B**), *B. animalis* subsp. *lactis* BB-12 (**C**) and *S. boulardii* CNCM I-745 (**D**) probiotic microorganisms. Solutions were produced with *L. ramosa* and *R. pusillus* partially purified β-galactosidase-active cocktails on lactose-fructose (red), skim milk (blue) or lactose (black) initial sugar substrates. The colony number counted in the absence of the given solution was taken as Control 1, while Control 2 contained the enzyme-free sample of the corresponding reaction mixture. Presented results are means of three replicates; error bars represent standard deviations. Different letters indicate statistical differences between the controls and the corresponding treatments (*p* < 0.05).

**Table 1 jof-07-00229-t001:** β-Galactosidase activity, cultivation temperature and origin of Mucoromycota isolates identified as the best producers in X-gal contained medium. The intensity of the blue color is proportional with the β-galactosidase activity.

Fungal Strains	β-Galactosidase Activity ^1^						Cultivation Temperature (°C)	Source
	*t*(incubation)/day							
	1	2	4	6	8	10		
*Lichtheimia ramosa* SZMC ^2^ 11360	+++	+++++	+++++	+++++	+++++	+++++	37	soil/unknown
*Lichtheimia corymbifera* SZMC 11361	++	++++	++++	+++++	+++++	+++++	37	soil/Afghanistan
*Lichtheimia hyalospora* SZMC 11364	++	++++	++++	+++++	+++++	+++++	37	*Manihot esculenta* stem / Ghana
*Rhizomucor miehei* SZMC 11005	++	+++	++++	++++	++++	+++++	37	peppermint compost/India
*Rhizomucor miehei* SZMC 11014	++	++++	+++++	+++++	+++++	+++++	37	compost/Switzerland
*Rhizomucor pusillus* SZMC 11025	+	+++	+++	++++	++++	+++++	37	dead fallen leaves/California, USA
*Rhizopus microsporus* var. *oligosporus* SZMC 13619	NC	+	+++	++++	++++	+++++	37	tempeh/Indonesia
*Mortierella echinosphaera* SZMC 11251	+	+	+++	++++	++++	+++++	20	begonia/Netherlands
*Umbelopsis longicollis* SZMC 11208	+	+++	++++	+++++	+++++	+++++	25	soil/Australia
*Umbelopsis ramanniana* var. *angulispora* SZMC 11234	+	+++	+++++	+++++	+++++	+++++	25	unknown/Russia

^1^ light blue color (+), darker blue color (++), blue color (+++); dark blue color (++++), deep dark blue color (+++++), NC: no blue color. ^2^ SZMC=Szeged Microbiological Collection.

**Table 2 jof-07-00229-t002:** Concentration and yield of oligosaccharides obtained with *L. ramosa* and *R. pusillus* β-galactosidase extracts (about 2400 units) on different initial materials.

Initial Materials	Crude β-Galactosidase Used	Oligosaccharide Concentration (mg/L) ^1^		Oligosaccharide Yield (%) ^2^
		Trisaccharide	Tetrasaccharide	
Lactose	*L. ramosa*	129.5 ± 10.7	34.5 ± 1.1	0.11
	*R. pusillus*	197.2 ± 2.5	60.4 ± 3.8	0.17
Skim milk	*L. ramosa*	14.4 ± 0.5	1.4 ± 0.03	0.032
	*R. pusillus*	20.1 ± 0.02	2.62 ± 0.02	0.047
Lactose–fructose	*L. ramosa*	88.6 ± 5.9	18.7 ± 0.4	0.11
	*R. pusillus*	187.6 ± 5.4	52.2 ± 3.1	0.24
*o*NPG–sucrose	*L. ramosa*	856.5 ± 16.5	48.1 ± 2.1	0.91
	*R. pusillus*	2,040.1 ± 36.8	196.2 ± 3.4	2.24

^1^ Reactions were carried out at 50 °C for 12 h in 50 mM of sodium acetate buffer (pH 6.0) contained 15% (*w/v*) lactose, 10% (*w/v*) skim milk, 10% (*w/v*) lactose +TABLE10% (*w/v*) fructose or 90 mM of *o*NPG + 10% (*w/v*) sucrose. Average values from three tests ± standard deviation. ^2^ Oligosaccharide yield was calculated using Equation (1) (see method Section 2.8).

**Table 3 jof-07-00229-t003:** Correlation coefficients (Pearson *r*) between growth-promoting activity towards different probiotics and total tri- or tetrasaccharide content achieved on each initial material after enzymatic treatments with crude β-galactosidase cocktails.

Initial Materials	*L. casei*	*L. acidophilus*	*B. animalis* subsp. *lactis*	*S. boulardii*
**Lactose**				
Trisaccharide	0.991	0.985	0.979	0.999
Tetrasaccharide	0.998	0.999	0.999	0.988
**Skim milk**				
Trisaccharide	0.791	0.995	0.994	0.961
Tetrasaccharide	0.635	0.992	0.993	0.875
**Lactose–fructose**				
Trisaccharide	0.785	0.986	0.999	0.926
Tetrasaccharide	0.698	0.957	0.989	0.869

**Table 4 jof-07-00229-t004:** β-Galactosidase production of Mucoromycota, applied fermentation conditions, growth substrates, and the enzyme activity achieved.

Mucoromycota Fungi	Fermentation Condition	Substrate	Enzyme Activity	Reference
*Mucor* sp.	SmF ^1^	lactose	228 U/L	[12]
*Rhizomucor* sp.	SmF	lactose	0.55 U/mL (0.21 U/mg)	[14]
	SSF ^2^	wheat bran	5.5 U/mL (2.04 U/mg)	
*Rhizomucor pusillus*	SmF	lactose	2.14 IU/mL	[16]
*Rhizomucor pusillus*	SSF	wheat bran	101.89 U/gds	[17]
*Rhizopus* sp.	SmF	lactose	less than 10 U/mL	[43]
*Rhizopus stolonifer*	SmF	lactose	2.250 IU	[28]

^1^ SmF: submerged fermentation. ^2^ SSF: solid-state fermentation.

## Data Availability

Not applicable.

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
