# Peer review of "β-Galactosidase-Producing Isolates in Mucoromycota: Screening, Enzyme Production, and Applications for Functional Oligosaccharide Synthesis"

_jof, 2021, doi:10.3390/jof7030229_

Round 1

Reviewer 1 Report

Dear authors,

  1. Do β-galactosidase of mucoromycota is different from the β-galactosidase of other species with same enzyme accession number? If not, why it has mentioned β-galactosidase of mucoromycota has rarely studied?
  2. Line 110: Twenty µl is inappropriate text. Should be changed.
  3. Line 118-126: what was the pH of the medium and did the pH of the medium got change upon progress of the time?
  4. Line 154: why it required to incubate 4-h at 4 degree Celsius.
  5. Line 168: 2400 U of enzyme was added or enzymatic activity was added?
  6. Growth of the microbes was done at 20, 25, and 37 degree Celsius, but enzymatic incubation was conducted at 50 degree Celsius. What is the logic of this?
  7. Line 185: 02 ml should be 200 µ
  8. Why electrospray ionization was operated in negative mode?
  9. Why Mortierella, Umbelopsis longicollis, and U. ramanniana was cultivated in 20, and 25 degree respectively while others were cultivated at 37 degree Celsius.

Regards

Reviewer 2 Report

  1. Wheat bran has been shown to significantly improve B-galactosidase production. Similar works on enzyme production using wheat bran should be added in the introduction.
  2. The source of organisms is not clearly stated. Was it isolated in this study or previously isolated and identified from other study? If isolated, describe the species identification process.
  3. Describe how spore suspension was prepared.
  4. How color intensity in plate screening was measured quantitatively? Measurement through naked eyes can be questionable. Any equipment used to aid the color detection such as colorimeter?
  5. Provide SDS-PAGE photos for each enzyme extract. SDS-PAGE must show that B-galactosidase was considerably major product of fermentation. Or else, specific activity in terms of U/mg is less accurate for a fair comparison.
  6. Any specific reason why only L. ramose and R. pusillus were proceed for GOS synthesis? Because in line 320-321 L. ramosa, L. hyalospora and R. miehei are reported to exhibit the highest increment of production.
  7. Provide SDS-PAGE of partial purified enzyme
  8. Only trisaccharide and tetrasaccharide are documented. What about other compounds in the product mixture such as galactose? How significant the GOS yield compared with other compounds?
  9. Different carbon source used (lactose, wheat bran and lactose, wheat bran) obviously would give different expression level as it contains different percentage of carbon that correlates with amount of enzyme produced. More explanation/discussion is needed on the significant enzyme expression in SmF with lactose-wheat bran and SSF with wheat bran. Support/compare with other studies from literature.
  10. Line 498-501, how much yield produced by these strains?
  11. Line 541-543, why it was shifted? It is worth to investigate before come out with this conclusion.
  12. Line 565 – 567, no evidence provided on any FFase/FTase produced. It is just an assumption.
  13. In probiotic growth experiment, GOS was not purified from its crude extract, how can we know the growth of probiotics are due to the GOS, not the other sugar or compounds contained in the extract?
